# Classification of Ataxic Gait

**DOI:** 10.3390/s21165576

**Published:** 2021-08-19

**Authors:** Oldřich Vyšata, Ondřej Ťupa, Aleš Procházka, Rafael Doležal, Pavel Cejnar, Aprajita Milind Bhorkar, Ondřej Dostál, Martin Vališ

**Affiliations:** 1Department of Neurology, Faculty of Medicine in Hradec Králové, Charles University, 500 03 Hradec Králové, Czech Republic; bhorkara@lfhk.cuni.cz (A.M.B.); onddost@gmail.com (O.D.); martin.valis@fnhk.cz (M.V.); 2Department of Computing and Control Engineering, University of Chemistry and Technology in Prague, 166 28 Praha 6, Czech Republic; tupa.ondrej@gmail.com (O.Ť.); A.Prochazka@ieee.org (A.P.); pavel.cejnar@vscht.cz (P.C.); 3Czech Institute of Informatics, Robotics and Cybernetics, Czech Technical University in Prague, 160 00 Prague 6, Czech Republic; 4Department of Chemistry, Faculty of Science, University of Hradec Králové, 500 03 Hradec Králové, Czech Republic; rafael.dolezal@gmail.com

**Keywords:** gait, ataxia, SARA, classification, machine learning

## Abstract

Gait disorders accompany a number of neurological and musculoskeletal disorders that significantly reduce the quality of life. Motion sensors enable high-quality modelling of gait stereotypes. However, they produce large volumes of data, the evaluation of which is a challenge. In this publication, we compare different data reduction methods and classification of reduced data for use in clinical practice. The best accuracy achieved between a group of healthy individuals and patients with ataxic gait extracted from the records of 43 participants (23 ataxic, 20 healthy), forming 418 segments of straight gait pattern, is 98% by random forest classifier preprocessed by t-distributed stochastic neighbour embedding.

## 1. Introduction

Motion disorders are a frequent manifestation of many diseases in neurology, rheumatology, orthopaedics or rehabilitation [1]. Approximately 70% of neurological inpatients show an abnormal gait [2]. The development of affordable technology capable of automatic recognition and further monitoring of pathological states is a method for improving disease management and reducing the load on the healthcare system. Tools that are specific for the neurological condition and clinical assessment are needed [3].

Motion analysis has a wide range of applications in these fields, allowing the detection of motion disorders, enabling an early diagnosis and further monitoring of patient state and treatment efficacy [4]. Wearable gait sensors have a significant correlation with the Scale for the Assessment and Rating of Ataxia (SARA) [5]. Even triaxial accelerometer-based smartphone applications showed a strong correlation with the score obtained with the clinical scales [6]. The values measured in the laboratory environment may differ significantly from the values measured during normal life activities. For example, the average daily gait speed was significantly lower than the average in-laboratory gait speed [7]. Motion sensors enable examination both in outpatient conditions and in the conditions of patients’ natural environment. The same motion sensors can be used in these patients, for example, to monitor cycling activities [8]. When monitoring walking outdoors, it is advisable to combine data from motion sensors with a global positioning system [9].

Human movement can be captured by different microelectromechanical sensor units (MEMS), video and depth camera systems, wireless communication links or pressure-sensitive walkways. Accelerometer sensors are generally more user friendly and less invasive than other sensors for the gait analysis [10]. Gait measured using wearables is generally more sensitive to group differences than instrumented walkways [11]. Inertial measurement units (IMU) made of MEMS usually consist of three-axis accelerometer sometimes accompanied by a magnetometer or gyroscopic sensor [12].

The Perception Neuron 2 (Noitom Ltd., Beijing, China) is a tool to deliver small, adaptive, versatile and affordable motion capture technology. The modular system is based on the neuron, an IMU composed of a 3-axis gyroskope, 3-axis accelerometer and 3-axis magnetometer. The strength of the system lies in the Perception Neuron’s 2 proprietary embeded data fusion, human body dynamics and physical engine algorithms which deliver smooth and true motion with minimal latency [13]. The Perception Neuron 2 consists the interconnected synchronized motion sensors based on a triaxial accelerometer, a gyroscopic sensor and a magnetometer. Depending on the number of sensors, the sampling frequency is 60 or 120 Hz. Scanned data can be stored on a micro-SD card in the device hub or can be sent via WiFi or bluetooth to an external device.

The large volume of data produced is a challenge for processing for clinical purposes. Reduction of the volume of stored data from motion sensors can be achieved, for example, by applying cubic splines [14]. Principal Component Analysis (PCA) and Linear Discriminant Analysis (LDA) are often used for feature extraction and dimensionality reduction. Karahoca et al. compared different methods of data reduction and classification. The best accuracy—93.8%—had combination Hu moments and k-nearest neighbours. Other methods used for the data reduction include PCA and LDA and methods for classification include support vector machines [15]. Those methods rely on the assumption that the data must lie approximately on a linear subspace of the high-dimensional data [16]. Mandery et al. [17] achieved an accuracy of 94.8% on their data set in four-dimensional feature space with dimensionality reduction method based on Hidden Markov Models. Publications testing discrete gait changes in multiple slerosis by shallow machine learning methods in combination with data dimensionality reduction methods were not found. Shallow and deep machine learning methods are better suited for classifying motion sensor data than statistical methods [18]. When processing a large amount of data obtained from motion sensors, parameterization or other data reduction methods are usually used. The feature extraction methods map the data from high-dimensional space to low-dimensional space and may use linear or non-linear mapping (Table 1). The reduced data is used to teach classifiers [19]. For shorter sections of recorded steps, it is more appropriate to use shallow learning methods after previous data reduction and parameterization. For example, in cerebellar ataxia, the amplitude of lateral deflections is an important parameter [20]. A significant reduction of acceleration parameters in neurological patients with ataxia and Parkinson’s disease was observed [21]. Machine learning methods have been successfully applied to recognise gait abnormalities in Friedrich’s ataxia [22], early-onset ataxia and developmental coordination disorder [23]. The principal component analysis of data from accelerometers was used to estimate the severity of ataxic gait quantitatively using a triaxial accelerometer [24]. Deep learning methods require long stretches of walking, which are difficult to obtain in patients with ataxia. The accuracy of the gait classification depends on the position of the sensors [25].

The purpose of this study is to compare accuracy of different methods of data reduction and shallow learning and to choose the most effective methods to use for classification of gait data from motion sensors in clinical practice. We hypothesise that the combination of random forest with the most modern method of data reduction uniform manifold approximation and projection will have the highest accuracy. In the following text, we first deal with the method of data acquisition, then data preprocessing, selection and description of their features and scaling and balancing. Next, we describe the methods of dimensionality reduction and data classification.

## 2. Materials and Methods

The study was conducted in accordance with the principles for human experimentation as defined in the Declaration of Helsinki and International Conference on Harmonization Good Clinical Practice guidelines and approved by the Ethics Committee of the University Hospital Hradec Kralove. Informed consent was obtained from each study participant after they had been told of the potential risks and benefits as well as the investigational nature of the study. The data processing procedure is schematically illustrated in the flow below (Figure 1).

### 2.1. Data Acquisition

The data was acquired using a Perception Neuron 2 device from 43 participants walking 100 m. The walk was performed on a connecting corridor in the hospital with a length of 50 m. Turning was excluded from the data processing. Of these, 23 patients had multiple sclerosis with ataxia and 20 healthy study participants served as controls. The mean age of the patients was 43.4±9.7 years, and the mean age of the control group was 35.9±15.0 years (*p* = 0.055). The mean body mass was 174.4±7.1 cm in the patients group and 172.9±6.8 cm in the control group(*p* = 0.485). The mean weight was 71.7±5.2 cm in the patients group and 74.3±4.3 kg in the control group (*p* = 0.084).

### 2.2. Data Preprocessing

The sampling frequency was 60 Hz or 120 Hz, depending on the number of sensors used (60 fps with 32 neurons, 120 fps with 17 neurons). The recordings taken at 120 Hz were subsampled to 60 Hz. The usual number of sensors used was 16. The records varied in length of the walked path and the number of repetitions depending on the severity of the patient’s disability. An algorithm for turn and step detection was developed to avoid this discrepancy among samples.

### 2.3. Feature Description

The shallow machine learning methods usually require parametrization of the recordings. Frequently used parameters of walking include gait velocity, cadence, step length, step regularity, step repeatability and the degree of body sway [26]. The coefficient of variance of stride length, smoothness and rhythm have the highest discriminatory value in the classification of gait disorders in cerebellar ataxia [27]. In neurologically ill patients, the symmetry of the disability is also important [28]. The following features were used:Step cadence is defined as a number of steps per unit time. In normal gait, cadence is approximately 100 to 115 steps per minute.The step length is defined as the Euclidean distance between adjacent foot contacts with the surface. There are several steps within each segment, and step length is defined as the median value from these distances.Step trajectory of one step is the length of a sampled curve between adjacent foot contacts with the surface. Trajectories are computed for each step and the median value for each segment to build a feature step trajectory.The relative step length as the ratio between step length (SL) and step trajectory (ST). This ratio attempts to reveal the optimality of a trajectory. Value closer to number 1, the minimum distance of the foot was travelled from point A to B, which should lead to the optimal step.Standard deviation of centre of mass (COM STD) trajectoryEnergy in frequency bands takes on input accelerometer data. The maximum valid frequency for all samples is 29 Hz; therefore, two energy bands were empirically chosen: [3, 15] Hz and [15, 29] Hz.

### 2.4. Feature Scaling

Different features may have different magnitudes, units and ranges. Some classification algorithms can adapt to the data with not normalised inputs, but the majority of them cannot. For example, if there is the Euclidean distance between two data points implemented in the algorithm, only the highest order feature will affect the overall classification results of this classifier and the others will have only a small effect on the classifier learning status. The following methods were used where appropriate:Rescaling. Synonym to rescaling is min-max normalisation; from each value the minimum is subtracted and divided by a range of values (the difference between the maximum and minimum values). The rescaled values lie within range [0, 1].Mean normalisation. The feature vector is normalised by subtracting the average value and divided by a range of values.Standardisation or z-score normalization is the most commonly used rescaling technique. Mean values are subtracted, followed by division by the standard deviation.

### 2.5. Class Balancing

To comply with the conditions of providing proper input data for applying various classification algorithms, we need to balance the number of segments by the class affiliation. The main reason for class balancing is maintaining the robustness and generality of classifier learning to avoid overfitting effects. The following methods were used for class balancing:Subsample majority class.Oversample minority class.Oversample by adding noise to the minority class.Oversampling by SMOTE (synthetic minority oversampling technique).

### 2.6. Dimensionality Reduction Techniques

Principal component analysis (PCA [29]). Linear dimensionality reduction using singular value decomposition (SVD) for data projection to a lower-dimensional space. The input data are centred but not scaled for each feature before applying the SVD.T-distributed stochastic neighbour embedding (TSNE [30]) is a technique for the dimensionality reduction that is particularly well suited for the visualisation of high-dimensional data sets.Uniform manifold approximation and projection (UMAP [31]) is a dimension reduction technique that can be used for visualisation similarly to t-SNE but also for general non-linear dimension reduction.

### 2.7. Classification

Logistic regression [32] is an extension of the linear regression model for classification. The difference against linear regression is the ability to perform classification on an accepted level of generality even for more complex data.Linear support vector machine [33]. Support vector machine algorithms attempt to find the optimal hyperplane in an N-dimensional space that distinctly classifies input data points. There are many possible hyperplanes to choose from, and the SVM algorithm selects hyperplane with the maximum margin (maximum distance between samples across all classes).Kernel support vector machine [34].Naive Bayes [35] used a probabilistic machine learning model for classification.A quadratic classifier [36] is a general version of the linear discriminant analysis (LDA). Unlike the LDA, it assumes that each class can have a different covariance matrix.k-nearest neighbour [37] can be used to solve both classification and regression problems. Implementation is easy, straightforward and understandable. The major drawback of becoming significantly slow as the size of the data in use increases. The basic idea is that similar things (data points) exist in close proximity.Decision tree [38] is a supervised learning algorithm. In the tree structure an internal node represents feature, the branch represents a decision rule, and each leaf node represents the outcome. This algorithm can be used for both prediction and classification.Random forest [39]. Decision trees have a problem with overfitting to the specific training set. The random forest is a combination of several decision trees. Decision rules given by individual trees are weighted and combined, and such methods, in general, are called ensemble methods.The neural network [40] (NN) is a computational model inspired by naive ideas about animals’ central nervous systems functions. NNs are capable of machine learning, pattern recognition, image compression or time-series prediction. The NN systems are composed of interconnected ”neurons” (real or hidden) that can compute values (weights) from patterns (inputs) by feeding (transfer functions) information through the network.AdaBoost [41] is an iterative ensemble method. It combines multiple weak learners, which are called decision stumps and can be used for both classification and regression problem.

### 2.8. Cross Validation

The principle of the cross-validation technique is to split the data into training and testing sets. For the training set 60% of the data was used and for the testing set, it was 40% of the data. The classifier is trained on the training set only, and for evaluation of accuracy, precision and other scores, the testing set is used. In addition, we used leave–one–group–out cross validation for our data. Hyperparameters of all models are available in Appendix A.

## 3. Results

Both different types of the dimensionality reduction techniques (Figure 2a,b and Figure 3a,b) and shallow classifiers (Figure 4) were compared.

### 3.1. Dimensionality Reduction Techniques

The compared methods of the data reduction show a different ability to discriminate ataxia and healthy controls as well as grouping according to the data of the study participants. Sensitivity, specificity and accuracy for the individual data reduction methods are given in Table 2 and Table 3.

### 3.2. Classification

Of all the data reduction and shallow learning methods tested, the random forest achieved the highest accuracy 98% in combination with TSNE data reduction. In combination with PCA, random forest reached 95%, while in combination with UMAP, it reached only 92% (Figure 4, Table 4).

## 4. Discussion

The aim of the study was to find a combination of a shallow classifier with the method of data reduction, which has the highest accuracy in the assessment of gait ataxia. The combination of TSNE to reduce the data evaluated by the random forest method seems to be the most appropriate procedure for classifying the gait recordings from motion sensors (Table 5).

The highest achieved accuracy of atactic gait classification in our study was 98%. Karahoca et al. obtained accuracy of 93.8% with combination Hu moments and k-nearest neighbours [15]. Mandery et al. [17] had comparable accuracy of 94.8% on their data set in four-dimensional feature space with dimensionality reduction method based on Hidden Markov Models. However, it is difficult to compare the accuracy of the classification of different degrees and types of the neurological gait disorders. The application of different methods of data reduction and classification on the same data set would bring comparable results.

Machine learning approaches are mainly applied to assess abnormal gait patterns due to their ability to work with multidimensional non-linear features [42]. The question is how ataxia affects the accuracy of gait parameter estimation. For example, analyses demonstrate that there is no correlation between the gait speed estimation error and impairment severity [43]. When classifying gait in patients with gait disorders, it is usually not possible to collect enough data to learn deep neural networks. In this case, the use of shallow learning methods is possible. The deep convolutional neural network can handle relatively small data sets [44]. Of the shallow learning methods, the random forest is good for a large number of problems. This method is based on trees; therefore, the scaling of the variables does not matter. The random subspace method and bagging are used to prevent overfitting. The automated feature selection was successfully used in the evaluation of cerebellar ataxia, where the key features were velocity irregularity and resonant frequency characteristics of truncal and lower limb movements [45]. The random forest was successfully used to evaluate which individual characteristics had the highest influence on the gait local dynamic stability [46]. The TSNE data reduction method in contrast with PCA retains non-linear variance. UMAP is similar to TSNE. The reason for the lower performance in this case may be UMAP’s inability to separate the nested clusters. Furthermore, TSNE can successfully separate subjects according to the severity of the clinical disability, as shown in Figure 5.

The torso and movements of the upper limbs are also involved in the walking stereotype. In the further studies, it is necessary to focus on the inclusion of parameters describing the compensatory movements of the torso and upper limbs during ataxic gait.

The limitation of the study is the relatively small number of study participants. The difference in age and weight in both groups of participants was just above the 5% level of significance and could affect the accuracy of the classification.

Binary classification is not of great importance for clinical practice or for pharmacological studies. However, T-distributed stochastic neighbour embedding is able to well separate subjects according to the severity of the clinical disability measured by the Scale for the Assessment and Rating of Ataxia score. This can be a good basis for multiclass classification using a random forest. This approach should already be used in both clinical practice and pharmacological studies.

Both clinical practice and expensive pharmacological studies in gait disorders are based on clinical impression and inaccurate clinical scales. Measurement and classification of gait disorders using motion sensors offers reproducible results that can refine the diagnosis and reduce the cost of testing new drugs by reducing the variance in the evaluation of the achieved results.

## Figures and Tables

**Figure 1 sensors-21-05576-f001:**
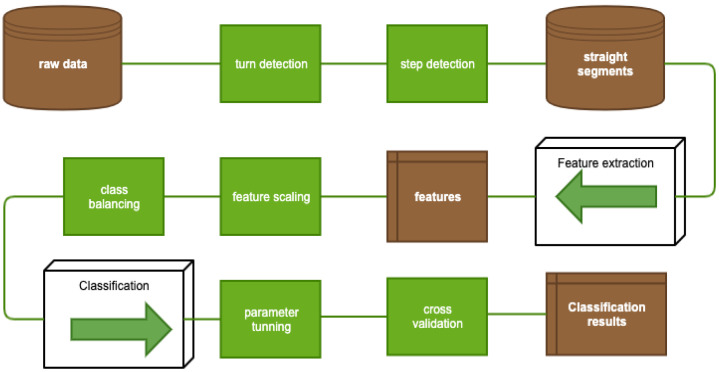
Flow diagram of data processing.

**Figure 2 sensors-21-05576-f002:**
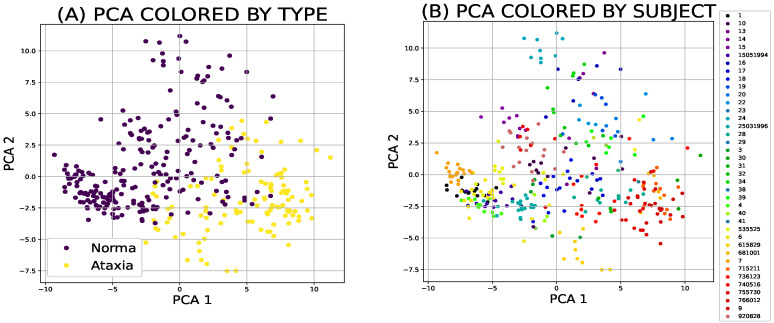
Principal component analysis is a tool that ranks the features by the eigenvalues, which corresponds to the desired ability of discrimination between classes (**A**). Of all the techniques compared, it has the worst ability to categorise individual participants (**B**).

**Figure 3 sensors-21-05576-f003:**
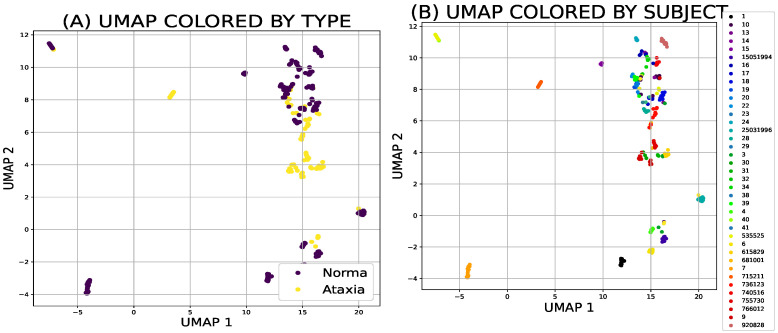
The uniform manifold approximation and projection model can be used for predicting a new sample to group (**A**) or similarity to the other study participant (**B**).

**Figure 4 sensors-21-05576-f004:**
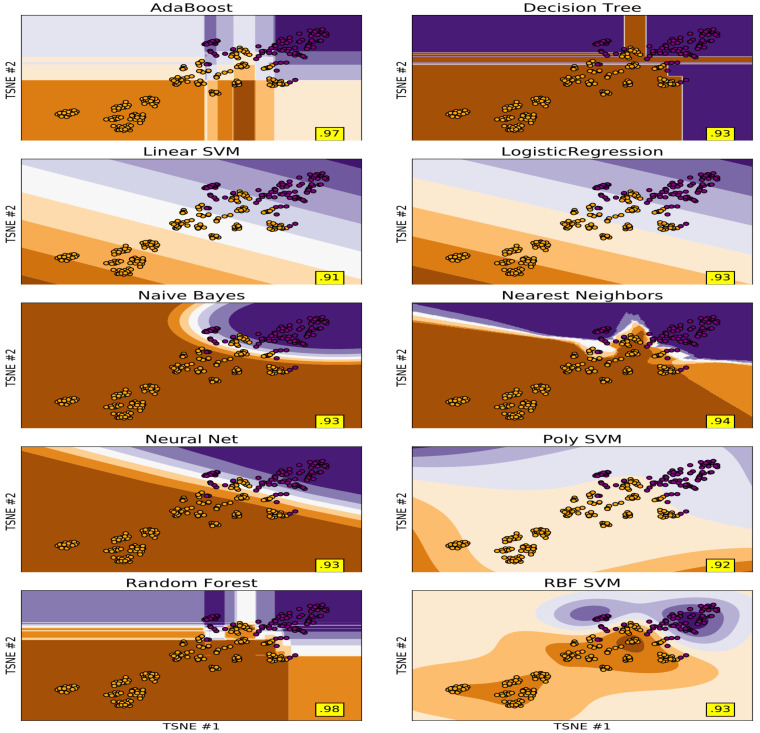
Classifier decision surface plotted for selected classifiers in dimension of two T-distributed stochastic neighbour embedding components. The achieved accuracy of classification for each specific case is shown in the yellow box in the right bottom corner.

**Figure 5 sensors-21-05576-f005:**
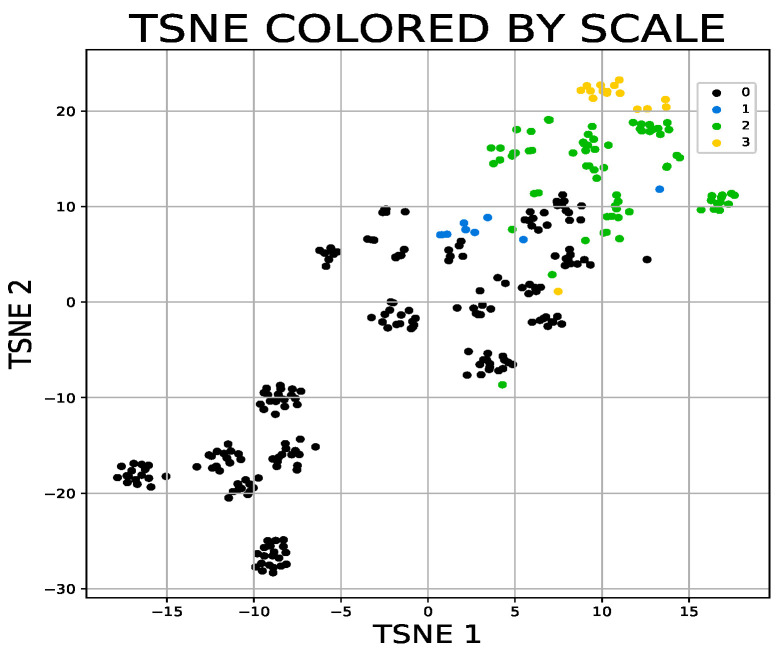
T-distributed stochastic neighbour embedding is also able to well separate subjects according to the severity of the clinical disability measured by the Scale for the Assessment and Rating of Ataxia score (the higher score the more severe ataxia).

**Table 1 sensors-21-05576-t001:** Common data reduction methods.

Method	Linear	Description
Principal component analysis	Yes	Adaptive exploratory method which creates new non-correlated variables that maximise variance
Non-negative matrix factorisation	Yes	Algorithm which decomposes one non-negative matrix into two smaller non-negative matrixes
Kernel principal component analysis	No	Kernel method projects not linearly separable data set into higher dimensional space where it is linearly separable, then PCA is applied
Linear discriminant analysis	Yes	This algorithm constructs he lower dimensional and minimises the within class variance
Generalised discriminant analysis	No	Kernel method projects not linearly separable data set into higher dimensional space where it is linearly separable, then GDA is applied
Autoencoder	No	Neural network architecture with bottleneck forcing a learned compression of the input data
T-distributed stochastic neighbour embedding	No	Visualisation method which keeps similar data points close together using heavy-tailed Student-t distribution to compute similarity between two data points
Uniform manifold approximation and projection	No	UMAP is used both for visualisation and for general non-linear dimension reduction. It is based on Riemannian geometry and algebraic topology.

**Table 2 sensors-21-05576-t002:** Accuracy, sensitivity and specificity for UMAP and all tested combinations of shallow learning methods.

Name	Accuracy	Sensitivity	Specificity
LogisticRegression	0.88	0.76	0.95
Linear SVM	0.87	0.76	0.94
Poly SVM	0.89	0.80	0.94
RBF SVM	0.94	0.84	1.00
Naive Bayes	0.87	0.84	0.89
Nearest Neighbours	0.94	0.90	0.96
Decision Tree	0.95	0.94	0.95
Random Forest	0.96	0.90	0.99
Neural Net	0.70	0.20	1.00
AdaBoost	0.96	0.88	1.00

**Table 3 sensors-21-05576-t003:** Accuracy, sensitivity and specificity for PCA and all tested combinations of the shallow learning methods.

Name	Accuracy	Sensitivity	Specificity
LogisticRegression	0.93	0.88	0.96
Linear SVM	0.90	0.73	1.00
Poly SVM	0.90	0.78	0.98
RBF SVM	0.91	0.88	0.93
Naive Bayes	0.93	0.84	0.99
Nearest Neighbours	0.93	0.90	0.95
Decision Tree	0.93	0.96	0.92
Random Forest	0.93	0.96	0.92
Neural Net	0.75	0.33	1.00
AdaBoost	0.94	0.96	0.93

**Table 4 sensors-21-05576-t004:** Accuracy for all tested combinations of shallow learning methods and data reduction.

Condition	PCA	TSNE	UMAP
AdaBoost	0.93	0.97	0.90
Decision Tree	0.93	0.93	0.90
Linear SVM	0.80	0.91	0.65
Logistic Regression	0.93	0.93	0.61
Naive Bayes	0.93	0.93	0.61
Nearest Neighbors	0.91	0.94	0.90
Neural Net	0.93	0.93	0.65
Poly SVM	0.91	0.92	0.65
Random Forest	0.95	0.98	0.92
RBF SVM	0.91	0.93	0.89

**Table 5 sensors-21-05576-t005:** Accuracy, sensitivity and specificity for TSNE and all tested combinations of the shallow learning methods.

Name	Accuracy	Sensitivity	Specificity
LogisticRegression	0.93	0.88	0.95
Linear SVM	0.91	0.76	1.00
Poly SVM	0.95	0.88	0.99
RBF SVM	0.95	0.88	0.99
Naive Bayes	0.90	0.90	0.90
Nearest Neighbours	0.96	0.92	0.98
Decision Tree	0.96	0.94	0.96
Random Forest	0.96	0.94	0.98
Neural Net	0.90	0.73	1.00
AdaBoost	0.95	0.92	0.96

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
