# Peer review of "Classification of Ataxic Gait"

_sensors, 2021, doi:10.3390/s21165576_

Round 1

Reviewer 1 Report

Although the work can be interesting, the manuscript is very little worked and have poor quality. General comments have been provided for the authors to improve it since it is difficult to evaluate the article in detail in its current state.

Introduction: The introduction section should be structured. In the current state, it is only one paragraph that exposes different ideas. Please, you should structure your text in paragraphs, and each paragraph should develop a main idea. The last paragraph should be the objective of the study. It will help the reader to follow the text easily. Right now, it looks like a chaotic text.

In the same way, many sentences should improve their writing. It is necessary to add connectors to enhance the meaning of the sentences. Please, review the introduction thoroughly and improve it by following these comments.

Introduction: Please, include references to support the statements: lines 11-12, 16-18, line 33, lines 36-39

Introduction: Authors should present the previous similar literature to show state of art and the novelty of their investigation

Introduction: Please, at the end, include a hypothesis of your study.

Material and methods, line 57: “All studies”. Are you using data from different investigations? Please, if this is the case, the text should be clarified.

Material and methods, lines 63-65: Different comments are provided to improve this section. “Perception Neuron” is the name of the device but not the type. Please, include the type of device, and add the data of the company within a parenthesis. Please, add more data to characterize the sample (body mass, height, etc.) and clarify if there are differences in these demographic data between groups. Avoid the use of the term “subject” due to ethical reasons. Include more information about the experimental procedure (have they performed a warm-up? Was the 100 m at maximum velocity or a comfortable velocity? How was space when they walked, in a treadmill or a room doing circles? Etc.

Material and methods, lines 126-153: Authors must do as best as possible to warranty the reproducibility of the analysis. Therefore, hyperparameter configuration should be reported for each classification method.

Material and methods, lines 155-157: Authors should report the percentage of data used for training and the percentage used for testing.

Material and methods: Authors should report which data they will use to assess performance classification. I recommend the authors include in table 1, the sensitivity, the specificity, and the area under the curve (AUC).

Figure captions and table captions. Please, include the full name of the abbreviations in the captions.

Discussion: The discussion is so short that it suggests whether the work presented should be a technical report rather than a research article. The authors did not report future studies, limitations, comparison of their results with similar previous studies, etc.

Author Response

1st reviewer

  1. Introduction is structured and extended. All changes are highlighted in red.
  2. The references to support the statements: lines 11-12, 16-18, line 33, lines 36-39 were added
  3. The previous similar literature to show state of art were included and the novelty of our investigation was explained
  4. Hypothesis of our study was included at the end of introduction
  5. Material and methods: The wording "all studies" has been changed to "The study"
  6. Material and methods, lines 63-65: Type of device was included. Demographics and statistical significance of the differences have been added. The term “subject” was replaced. More information about the experimental procedure were included. 
  7. Material and methods, lines 126-153: The hyperparameter files are too large to include in the article. They are attached as an appendix to the article.
  8. The percentage of data used for training and the percentage used for testing are reported
  9.  The sensitivity and the specificity are reported. Area of ROC curve can not be calculated from our data
  10.  The full name of abbreviations were included in figure and tabel captions
  11. Research article was changed to communication

Reviewer 2 Report

The paper explores different strategies to reduce the data size in clinical studies. Numerical validations demonstrate the importance of this study.

Some comments to improve this research are the following:

  1. The introduction section can be improved by incorporating a Table that summarizes the main approaches reported in the literature to reduce the data size.
  2. At the end of the Introduction, please include a paragraph that explains the document's organization.
  3. Section 3 requires to be more details in its presentation. The authors omit multiple details on it. Please revise it. 
  4. A flow diagram that presents the main steps to implement all the studied methods can help the readers to easily understand your work.

The paper is very interesting; however, it must be reconsidered for the second round of review.

Author Response

  1. The table that summarizes the main approaches reported in the literature to reduce the data size was incorporated
  2. At the end of the Introduction, have been included a paragraph that explains the document's organization
  3. Section 3 was revisited and extended
  4. A flow diagram that presents the main steps to implement all the studied methods was included.

Round 2

Reviewer 1 Report

The manuscript was considerably improved by the authors. However, there are different aspects that should be improved:

-Please, review carefully the manuscript to find typos (e.g., line 86, line 115, line 117, line 158, etc.) and grammar mistakes (e.g., line 181).

-Line 90, modify “weight” by “body mass”

The authors did not improve the discussion section which makes the manuscript unsuitable for publication. Please, include a first paragraph with the aim of the study and the main results. Improve the discussion section comparing your results with similar previous studies, add a paragraph with a proposal of future studies, add a paragraph with the limitations of your study, add a paragraph with the practical applications of your study, and a final paragraph with the main conclusion of your study.

Author Response

Typos on line 86 was not found. Other typos and grammar mistakes have been fixed. All required paragraphs were included in the discussion. Changes are in red.

Reviewer 2 Report

I appreciate the effort of the authors to improve the quality of the manuscript. This is now suitable for publication. No further comments. 

Good job.

Author Response

Typos and grammar mistakes have been fixed.